# Investigations on the Optimal Estimation of Speech Envelopes for the Two-Stage Speech Enhancement

**DOI:** 10.3390/s23146438

**Published:** 2023-07-16

**Authors:** Yanjue Song, Nilesh Madhu

**Affiliations:** IDLab, Ghent University—imec, 9000 Gent, Belgium; nilesh.madhu@ugent.be

**Keywords:** speech enhancement, speech envelope estimation, GRU, CRNN

## Abstract

Using the source-filter model of speech production, clean speech signals can be decomposed into an excitation component and an envelope component that is related to the phoneme being uttered. Therefore, restoring the envelope of degraded speech during speech enhancement can improve the intelligibility and quality of output. As the number of phonemes in spoken speech is limited, they can be adequately represented by a correspondingly limited number of envelopes. This can be exploited to improve the estimation of speech envelopes from a degraded signal in a data-driven manner. The improved envelopes are then used in a second stage to refine the final speech estimate. Envelopes are typically derived from the linear prediction coefficients (LPCs) or from the cepstral coefficients (CCs). The improved envelope is obtained either by mapping the degraded envelope onto pre-trained codebooks (classification approach) or by directly estimating it from the degraded envelope (regression approach). In this work, we first investigate the optimal features for envelope representation and codebook generation by a series of oracle tests. We demonstrate that CCs provide better envelope representation compared to using the LPCs. Further, we demonstrate that a unified speech codebook is advantageous compared to the typical codebook that manually splits speech and silence as separate entries. Next, we investigate low-complexity neural network architectures to map degraded envelopes to the optimal codebook entry in practical systems. We confirm that simple recurrent neural networks yield good performance with a low complexity and number of parameters. We also demonstrate that with a careful choice of the feature and architecture, a regression approach can further improve the performance at a lower computational cost. However, as also seen from the oracle tests, the benefit of the two-stage framework is now chiefly limited by the statistical noise floor estimate, leading to only a limited improvement in extremely adverse conditions. This highlights the need for further research on joint estimation of speech and noise for optimum enhancement.

## 1. Introduction

Speech captured by microphone in the real-world environment is prone to being corrupted by background noise. In order to reduce listener fatigue and the loss of intelligibility, speech enhancement which aims at removing the background noise and improving intelligibility has been an important and active field for many years.

The established statistical speech enhancement methods, the minimum mean-square error (MMSE), short-time spectral amplitude (STSA) estimator [1], and the MMSE-log spectral amplitude (MMSE-LSA) [2], require an initial estimate of a priori SNR and a posteriori SNR derived from the spectral density power estimates of the clean speech and the background noise. On top of that, the decision-directed (DD) approach—a recursive smoothing procedure for the a priori SNR—is proposed in combination with these estimators to reduce the residual musical tones and improve the naturalness of the processed audio. However, this technique also introduces an estimation bias in the SNRs and leads to an annoying reverberation effect [3]. Therefore, a two-stage framework has been proposed in Reference [3] to avoid the speech distortion from this bias. In the two-stage framework, a better speech estimate is refined from the initial one so that a priori SNR can be calculated *without* recursive smoothing. Additional prior knowledge on speech can also be introduced during the procedure.

In our recent work [4], for example, we improve the speech harmonic recovery method termed cepstral excitation manipulation (CEM) [5] using the source-filter model of speech production to highlight its periodic structure. In this model, the speech signal is decomposed into an excitation and an envelope component in order to represent the excitation source and the vocal tract filter, respectively. It is proposed to amplify the quefrency related to the fundamental frequency and its harmonics in order to highlight the periodic structure of the voiced speech in the cepstral domain. To maintain the fine structure of the speech, the excitation signal in high quefrencies is smoothed with a quefrency-dependent window. In this work, we investigate the contribution of enhancing the other component of the source-filter decomposition result, the spectral envelope, for speech enhancement.

It has been shown that there is a strong correlation between the speech envelope and its intelligibility [6]. Consequently, the short-time spectral envelope of speech has been widely exploited in many areas such as automatic speech recognition (ASR) [7], artificial bandwidth extension [8], and speech intelligibility prediction [6,7]. The well-known and widely used short-time objective intelligibility (STOI) [9] is based on the linear correlation between the envelopes of the processed noisy speech signal and those of the clean reference. One well-known problem of speech enhancement is that many methods hardly improve (and, sometimes, even degrade) speech intelligibility, although they perform well on noise reduction. Given the relationship between the speech envelope and intelligibility, it is possible to improve speech quality as well as its intelligibility by refining the spectral envelope of the clean speech estimate.

For a given language, the possible patterns of speech spectral envelope are limited because the number of phonemes is limited, which makes the codebook technique an efficient solution to capture a priori information about speech envelopes. Thereby, a good estimate of the codebook entries integrates this prior knowledge into the following tasks. For example, [10] trained two sets of codebooks for speech and noise, respectively. Then, the gain function was estimated by the codebook-constrained Wiener filter, with optimal codebook entries being searched for in a maximum likelihood framework.

As speech has temporal dependency, a combination of the Gaussian mixture model (GMM) and hidden Markov model (HMM), GMM-HMM, was widely used in classical ASR systems as the back-end to recognise phonemes [7]. This statistical approach models the distribution of phonemes and their temporal dependency through two individual components: the GMMs, which learn the feature distribution, and the HMM, which imposes temporal dependencies on the hidden state sequences inferred from the GMMs. A similar pre-trained codebook with a GMM-HMM back-end also serves as the baseline in the speech envelope enhancement research of Reference [11] using the aforementioned two-stage framework. Therein, only the speech envelope codebook is generated, and the speech envelope is estimated from it in an MMSE manner. This envelope is introduced to update the a priori SNR for a second-stage estimate of the clean speech. In Reference [12], noisy signals are enhanced by resynthesising the clean speech from the inferred acoustic cues (e.g., pitch and spectral envelope). The underlying clean speech envelope is, again, estimated with a codebook-aided Kalman filter, the codebook having been designed to capture not only the envelope shapes, but also the evolution of the envelopes in a given number of consecutive frames.

The classifier for such codebook-based methods can, of course, be replaced by deep learning models. In Reference [13], using two separate sets of codebooks for speech and noise, the codebook entries corresponding to the envelopes of both components are estimated by a feedforward deep neural network (DNN). These codebook entries are used to update the time-smoothed Wiener filter which performs the final speech enhancement. The work in Reference [11] also investigates the utilisation of DNN-based classifiers for codebook-based speech envelope estimation. Compared to the GMM-HMM baseline, the trained DNN classifier, with a similar computational cost, shows an advantage in both the classification accuracy and the instrumental metrics for speech enhancement. Compared to its regression counterpart, in which the network architecture is kept but the model is repurposed to predict the envelope coefficients directly from the initial estimation, it is shown that this architecture benefits from the codebook.

There are different ways to extract and represent speech envelope. Subband representation is a popular approach. For example, in STOI, the spectro-temporal envelope is calculated as the average of one-third octave decomposition results of 30 consecutive frames [9]. The STOI loss function to optimise speech enhancement DNN adopts, naturally, the same features [14]. Analogously, Reference [15] uses equivalent rectangular bandwidth (ERB) to compress the spectrum and Reference [16] uses the auditory filterbank. It should be noted there is no direct inverse from these subband presentations to spectra. Applying the subband gain directly to the spectrum yields a ‘rougher’ signal, as the processed signal is less harmonic [15]. Therefore, the subband gain function is combined with a comb filter to restore the distorted harmonics in Reference [15].

The envelope can also be obtained from the auto-regressive (AR) filter applied, e.g., in linear predictive coding (LPC). For stability reasons, instead of directly using the AR filter coefficients, the equivalent line spectral frequencies (LSF) are adopted for speech enhancement in References [12,13]. Another equivalent representation of the AR filter coefficients is given by linear prediction cepstral coefficients (LPCCs). These are employed in Reference [11] to define the spectral codebook and to estimate the enhanced envelope within the two-stage framework. However, LPCCs can suffer from quantisation issues [17], which can cause degradation in codebook-based approaches. Another alternative envelope representation is based on the cepstral representation of the signal, which also implicitly describes the spectral envelope. Based on the relationship between the spectrum and the cepstrum, the first few cepstral coefficients (CCs) of a signal frame can be regarded as the description of its spectral shape. This is exploited in cepstral smoothing approaches [18,19,20] in order to remove musical noise. By preserving the first few cepstral coefficients, and strongly smoothing the rest, instantaneous temporal spectral fluctuations in the signal are suppressed while the principal structure (i.e., the spectral envelope) of the processed speech is maintained. Of the aforementioned envelope representations, we focus on the LPCCs and CCs in this work due to the convenience of their transformations between the domains and the easy fitting of the decomposition results to the source-filter model.

While data-driven, deep-learning-based end-to-end speech enhancement offers a powerful solution, the computational cost of such a system is still relatively high. Furthermore, a drawback of such systems is the black-box nature of the enhancement, which makes interpretability and control difficult. Data-driven envelope estimation, incorporated into classical speech enhancement, can provide us a compromise, and at a low computational complexity. This work is developed from the idea of cepstral envelope estimation (CEE) using the pre-trained codebook in [11]. We further explore its potential and investigate the achievable results of this method. Specifically, the following questions will be answered by our investigation: what is the maximum benefit of such data-driven two-stage enhancement? How (much) does the quantisation of the envelopes affect the quality of the enhanced audio quality? What is the optimal cepstral speech envelope representation for the purpose of speech enhancement? Will the speech envelope classifier benefit from temporal modeling?

Below, we start from a series of oracle tests to investigate the potential and the limitation of the codebook method, in which different envelope representations are compared and benchmarked against each other in the two-stage framework. Then, several practical systems are trained and evaluated.

The temporal dependency of speech is usually taken into consideration in the aforementioned envelope enhancement methods via explicit temporal models or components such as Kalman filters and HMMs in the frameworks. For the DNN structure, however, including an HMM is counterproductive, as reported in Reference [11]. Therefore, we will investigate other possibilities to enable temporal modeling within the neural network. Literature shows that recurrent layers are powerful to this end. For instance, the long-short term memory (LSTM) layer is widely used in end-to-end speech enhancement systems due to its effective usage of long-term information granted by the gate mechanism. Recent research [21] shows that gated recurrent units (GRU) can achieve comparable performance with less complexity, which is also verified in the speech enhancement tasks [22]. Therefore, in this contribution, we will investigate the performance of the GRU-based classifier for the speech envelope estimation using codebooks.

Furthermore, we will explore the usage of the more recent network architecture, convolutional recurrent neural network (CRNN), as the regression estimator. It is hypothesised in Reference [11] that the repurposed feedforward DNN is too small for the regression problem. Yet, fully connected neural networks have been gradually replaced by the convolutional layers and convolutional neural networks (CNN) have been reported to yield high performance on many tasks and to do so with fewer parameters than feedforward DNNs. For a deep or complex network, CNNs can be easily trained in an end-to-end style. By inserting the recurrent layers into CNN, the network benefits from both the strong feature extraction ability of convolutional layers and the temporal modeling ability of the recurrent layers. Therefore, we propose to make use of the CRNN architecture for the regression problem.

The remainder of this paper is organised as follows. We provide an overview of the two-stage speech enhancement framework in Section 2, so that the purpose and the target of CEE are clear. Section 3 introduces the cepstral envelope estimation in a systematic manner, followed by its use in the two-stage enhancement framework. We report and discuss the evaluation results of the oracle tests and the practical systems in Section 4 and answer the core questions raised above. The paper is summarised and concluded in Section 5.

## 2. Speech Enhancement Framework

We consider the noisy observation y(k), which consists of the target speech s(k) corrupted by noise v(k) in an additive way: y(k)=s(k)+v(k), with *k* being the discrete time sample index. The microphone signal can then be transformed using the short-time Fourier transformation (STFT) with an *M*-point windowed discrete Fourier transform (DFT). This yields $Y_{l}\left(m\right)=S_{l}\left(m\right)+V_{l}\left(m\right)$, where *m* is the frequency bin index and *l* is the frame index.

As summarised in Figure 1, we adopt the same two-stage speech enhancement framework in Reference [11]. A preliminary denoising is performed in the first stage. The MMSE-LSA gain function is employed to obtain the initial speech estimate ${\hat{S}}_{l}\left(m\right)$. As with the majority of the gain functions, this estimator ${\hat{G}}_{l}\left(m\right)$ is expressed as a function of two crucial parameters: a priori SNR $\xi_{l}\left(m\right)$ and a posteriori SNR $\gamma_{l}\left(m\right)$. They are defined as:(1)$$\xi_{l}\left(m\right)=\frac{\lambda_{s,l}\left(m\right)}{\lambda_{v,l}\left(m\right)}\;,$$
and
(2)$$\gamma_{l}\left(m\right)=\frac{|Y_{l}{\left(m\right)|}^{2}}{\lambda_{v,l}\left(m\right)}\;,$$
where $\lambda_{s,l}\left(m\right)$ and $\lambda_{v,l}\left(m\right)$ are the power spectral densities (PSDs) of the speech and noise signals, respectively. Since the true values of these PSDs cannot be obtained in practice, ${\hat{\gamma }}_{l}\left(m\right)$ is approximated using the estimated noise PSD ${\hat{\lambda }}_{v,l}\left(m\right)$ from the noise floor estimator, and ${\hat{\xi }}_{l}\left(m\right)$ is obtained from the decision-directed (DD) approach. The clean speech amplitude estimate is then obtained by applying the gain function to the amplitude of the noisy observation:(3)$$|{\hat{S}}_{l}\left(m\right)|=|Y_{l}\left(m\right)|\cdot {\hat{G}}_{l}\left(m\right).$$

Then, according to the source-filter model, the enhanced signal is decomposed into the excitation signal ${\hat{R}}_{l}\left(m\right)$ and the envelope ${\hat{H}}_{l}\left(m\right)$, and each component can be enhanced individually. The enhancement of the speech excitation signal has been discussed in References [4,5,23], showing that the idealised excitation signal ${\bar{R}}_{l}\left(m\right)$ brings the benefit of recovering the weak or lost harmonics in the initial speech estimate.

While the excitation signal can be modeled by straightforward mathematical equations due to its periodic nature in the voiced frames with the largest energy [4,5,23], data-driven methods are more common in the estimation of the speech envelopes as in References [10,11,12,13]. If the underlying clean-speech envelope can be accurately estimated from the distorted or noisy signal envelope, it should improve the final speech estimate. One option to introduce prior knowledge of speech envelopes is to use codebooks. Thereby, the envelope estimation problem is converted into a classification problem. First, we create a codebook representing the different speech envelope patterns. Next, we train a suitable classifier to estimate the correct codebook entry for each frame, conditioned on the initial estimate ${\hat{H}}_{l}\left(m\right)$. The other perspective is to regard the envelope estimation problem as a regression, which predicts the underlying clean-speech envelope from the noisy observation.

The improved speech envelope ${\bar{H}}_{l}\left(m\right)$ is subsequently combined with the refined speech excitation signal ${\bar{R}}_{l}\left(m\right)$, yielding an improved speech estimate ${\bar{S}}_{l}\left(m\right)$. However, this synthetic speech estimate sounds less natural than the initial speech estimate, as the excitation signal and the envelope are artificially imposed. Thus, instead of using ${\bar{S}}_{l}\left(m\right)$ to recover the speech, we use ${\bar{S}}_{l}\left(m\right)$ to update the a priori SNR:(4)$${\tilde{\xi }}_{l}=\frac{|{\bar{S}}_{l}{\left(m\right)|}^{2}}{{\hat{\lambda }}_{v,l}\left(m\right)}.$$

This is then used to compute a new gain function: ${\tilde{G}}_{l}\left(m\right)=g^{\mathrm{LSA}}\left({\tilde{\xi }}_{l}\left(m\right),{\hat{\gamma }}_{l}\left(m\right)\right)$. The final speech estimate ${\tilde{S}}_{l}\left(m\right)$ is given by applying this new gain ${\tilde{G}}_{l}\left(m\right)$ to the microphone signal $Y_{l}\left(m\right)$:(5)$${\tilde{S}}_{l}\left(m\right)=Y_{l}\left(m\right)\cdot {\tilde{G}}_{l}\left(m\right)\;.$$

The enhanced time-domain signal can be obtained from ${\tilde{S}}_{l}$ by over-lap add.

## 3. Cepstral Envelope Estimation

Since the envelope can be compactly represented in the cepstral domain, this estimation is named cepstral envelope estimation (CEE) in Reference [11]. It has been shown that the classification DNN (C-DNN) is the optimal system in their framework in comparison with the GMM-HMM baseline, DNN-HMM pipeline, and the regression DNN. Thus, we take C-DNN as the baseline of our research.

In this baseline system, the envelope is extracted by LPC analysis and represented by LPCCs. The codebook is generated in two steps. First, using the energy level criterion, the windowed frames of clean signals are divided into two categories, namely speech active frames and speech inactive frames. Then, the zero-mean speech active frames are clustered into *C* classes by the Linde–Buzo–Gray (LBG) algorithm [24]. To complete the codebook generation, one spectrally flat envelope is added as the template for silent frames. Once this codebook is generated, the speech active frames of the training data are labelled by assigning them to the closest template (codebook vector). A classifier can be trained using these labels and appropriate multi-condition data. During inference, the output of the classifier is interpreted as the posterior distribution of the codebook entries conditioned on the observation. The final envelope estimate ${\bar{c}}_{l}$ is then obtained either by maximum a posteriori considerations, or by a weighted sum of the different templates (i.e., the optimal estimate in the MMSE sense):(6)$${\bar{c}}_{l}=\sum_{i=0}^{C-1}p_{l}^{i}\cdot h^{i}\;,$$
where $h^{i}$ is the *i*th template in the codebook, and $p_{l}^{i}$ is the posterior probability of the *i*th template for frame *l*, conditioned on the noisy observation.

In this section, we will take a closer look at each individual step of this baseline to further optimise it for speech enhancement.

### 3.1. Feature Extraction

#### 3.1.1. LPCC

The LPC coefficients $\left(\left\{a_{l}\left(1\right),a_{l}\left(2\right),\ldots ,a_{l}\left(N\right)\right\}\right)$ of the AR model, for frame *l*, can be derived from the auto-correlation function [25] of the preliminary speech estimate. The coefficients are then converted to the cepstrum in the following recursive manner:(7)$$\begin{array}{cc} c_{l}\left(0\right) & =\ln N\\ c_{l}\left(p\right) & =a_{l}\left(p\right)+\sum_{i=1}^{p-1}\frac{i}{p}c_{l}\left(i\right)a_{l}\left(p-i\right)\;,\mathrm{for}\;1\leq p\leq N\;. \end{array}$$

These coefficients $c_{l}=\left\{c_{l}\left(1\right),c_{l}\left(2\right),\ldots ,c_{l}\left(N\right)\right\}$ derived from LPC are taken as speech envelope representations.

#### 3.1.2. Cepstral Coefficients (CC)

Cepstral coefficients are straightforward to calculate from the preliminary speech spectrum ${\hat{S}}_{l}\left(m\right)$ by a *M*-point iDFT as:(8)$$d_{l}\left(q\right)=\mathrm{iDFT}\left\{\log|{\hat{S}}_{l}\left(m\right)|\right\}\;,$$
with *q* being the quefrency bin. Given the symmetric nature of the cepstrum (property of the (i) DFT on real-valued symmetric spectra), only the first half of the cepstrum (from bin 0 to bin M/2) needs to be preserved for further investigation. Then, the coefficients can be divided into three parts according to the source-filter model: first, the energy term $d_{l}\left(0\right)$; next, the initial few coefficients representing the speech envelope, namely $d_{l}=\left\{d_{l}\left(1\right),d_{l}\left(2\right),\ldots ,d_{l}\left(N\right)\right\}$; and, lastly, the remaining coefficients encoding the speech fine structure.

### 3.2. Codebook

It is proposed to create the codebook from speech active frames in Reference [11]. The partition is reasonable given the purpose of the codebook, but it should be noted that the energy criterion is not perfect to generate speech activity detection labels. The short-time Fourier representation is computed on overlapped, windowed frames. Thus, some frames that are classified as speech inactive actually possess very weak speech (due to the leakage of speech into the adjacent silent frames), and thus, their envelopes move away from a flat shape. This indicates potentially more variance even in the *low-energy* frames. In CEE, this error can degrade the system performance in two possible ways. First, according to this procedure, the idealised flat envelope is assigned to all low-energy frames, although some of them are closer to one of the speech templates. If the classifier is perfectly accurate, this would introduce speech distortion to the speech estimate when updating the a priori SNRs according to (Equation 4). In addition, this assignment error increases the difficulty of classifier training. The codebook assignment can be regarded as a quantisation of the clean speech signal envelope, and the classifier is trained to learn the mapping from the distorted coefficients to these quantised templates. This is the standard setting of the classification problem. However, the frames are assigned based on different rules, which makes the learning target ambiguous: it can be either a mapping to the most-likely speech envelope template for a speech active frame, or a replacement by a complete flat envelope for a *low-energy* frame. Moreover, the major indication of this speech/non-speech mapping, the frame energy level, is not available to the classifier. From the point of view of network training, this manually separated ‘silent’ codebook entry actually introduces noise into the training set. Consequently, the network training could be deteriorated by this elaborate division.

In order to investigate the influence of this assignment error, we followed the procedure to create the LPCC codebook from the speech-active frames of the TIMIT training set. After including the ideal flat envelope for silence, we reassigned all the ‘non-speech’ frames to entries of this codebook based on the cepstral distance. Figure 2 depicts the codebook entry distribution for the two types of frames, where it is seen that, in fact, a large portion of the ‘non-speech’ frames have envelopes similar to templates corresponding to speech-active frames and, therefore, should *not* be quantised as a single idealised entry with a flat envelope. Another interesting observation from this figure is that the envelopes of these low-energy frames concentrate on a few entries.

In order to take a closer look at those wrongly assigned frames, we plot all the envelope templates in Figure 3 and arrange them in a descending order of the posterior distribution of the codebook entries on non-speech frames ($p(i|H_{0})$). In other words, the low-energy frames are more likely to take the envelopes on the left side of the figure. Two points are now obvious: (1) despite manually removing the low-energy (‘silence’) frames, the speech codebook can still contain spectrally flat templates; (2) low-energy frames have, more often than not, non-flat spectral envelope shapes.

With this observation on the existing codebook generating method, we propose to create the codebook from the envelopes of *all* frames in the clean-speech data. Figure 2 has shown that the clustering generates clearly distinct templates. Thus, separating the frames in advance by the energy criterion is not necessary. Using zero-mean LPCCs of all frames, now, we create a new *unified* codebook for the same speech corpus and plot the entry distribution in Figure 4. It can now be observed that, in contrast to using the separate codebook, the two types of frames are relatively mutually exclusive with regard to their distribution among the unified codebook entries. Therefore, the unified codebook could be a better choice for the speech envelope enhancement task. It should be noted that although we only demonstrate the comparison between the separate and the unified codebook for LPCC, an identical trend can also be observed in the CC-based codebooks.

### 3.3. Envelope Estimator

#### 3.3.1. Feedforward DNN Classifier

In Reference [11], it was shown that using a feedforward DNN for the classifier outperformed the GMM-HMM approach. This network has a stack of fully connected layers as hidden layers, and the output is normalised by the softmax layer for the classification. The investigation showed that when the size of the network is fixed, the number of hidden layers and the choice of activation functions both have very limited influence on the network classification accuracy. We take the network composed of four hidden layers as our baseline, because this network achieves the highest accuracy on both the development set and the test set. We choose the activation functions of the network as follows: Leaky ReLU for the input and hidden layers, and the sigmoid function for the output layer, followed by a softmax layer to normalise the output.

#### 3.3.2. Recurrent Neural Network-Based Classifier

Although the aforementioned DNN shows superiority to the GMM-HMM back-end baseline in terms of classification accuracy, the temporal modeling ability provided by HMM is missing in this DNN architecture. As a remedy, the DNN was complemented by an HMM. However, the HMM results in a performance bottleneck [11]. In this regard, recurrent neural networks could be a more suitable comparison to the GMM-HMM baseline. In order to examine the function of the recurrent layers in the envelope estimation, we investigate the GRU-based classifier in this work. We adopt the simplest architecture here: one or several GRU layers in cascade with one FC layer to compress the output feature dimension. The final output is normalised by the softmax function, allowing for its interpretation as the *a posteriori* probability distribution across the codebook entries.

#### 3.3.3. CRNN-Based Envelope Estimation by Regression

The envelope estimation problem can also be formulated as a regression problem from the distorted envelope coefficients to the clean ones. Yet, according to Reference [11], the performance of a regression feedforward DNN is imbalanced at different SNRs. We propose to use an alternative architecture, the convolutional neural network, for the regression problem. CNN is popular because of its self-learning feature extraction ability. In order to fully exploit this feature, we reformulate the original coefficient-to-coefficient envelope estimation problem into a regression from the noisy spectrum with the initial gain function estimate to the clean speech envelope coefficients. The two available features, the logarithm of the zero-mean noisy magnitude spectrum and the logarithm of the LSA gain function, are taken as two separate channels of the network input. After several convolutional layers with the leaky ReLU activation function, the feature map is flattened and combined by the FC layer. One GRU layer is employed as the final output layer of this regression network in order to combine the past states with the current prediction and obtain a final estimate of the envelope coefficients.

## 4. Evaluations

We will now evaluate the proposed speech envelope enhancement method in two settings: (a) the oracle tests that assume the classifier is perfectly accurate—this provides us with the basis for feature selection and demonstrates the full potential of the current framework—and (b) the practical system tests that evaluate and compare the different envelope estimation approaches in realistic settings.

### 4.1. Experimental Setup

All of the networks were trained and tested with the same synthesised data set; 90% of the TIMIT training set and 21 files from ETSI noise set were mixed at 6 SNRs: {−5 dB, 0 dB, 5 dB, 10 dB, 15 dB, 20 dB}. The remaining 10% of the TIMIT training set was reserved for the validation set. For the evaluation, the test set was created from the TIMIT test set and the unseen noise signals from the ETSI database (Car, Traffic) and QUT database (Cafe, Kitchen, and City) at the same 6 SNR levels. All of the speech and noise signals were down-sampled to 16kHz and high-pass filtered by a second-order Butterworth filter with a cutoff frequency at 100Hz before mixing. The SNRs for the mixing were calculated according to Reference [26] in which, for speech, the speech active level is used, and the noise level is computed using the long-term root-mean-square.

For all tests, the input noisy signal was first processed as follows in the preliminary denoising stage. A pre-emphasis filter with a coefficient of −0.97 was first applied. The signal was then segmented with 50% overlap and windowed by the square-root von Hann window of M=512 prior to computing its spectrum. For the LSA gain function, the smoothing factor of the DD approach was α=0.97. Further, the a priori SNR and a posteriori SNR were bounded between −40 and 40dB in order to avoid numerical issues. The gain function was, finally, lower-bounded to −15dB. The noise floor was estimated by the speech presence probability minimum mean-square error (SPP-MMSE) approach with fixed priors [27]. Without prior knowledge, we assumed an equal a priori probability for speech presence and absence as suggested, and the optimal a priori SPP was set to 15dB.

We kept the length of both feature vectors (LPCCs and CCs) set to N=20 for a fair comparison. The sizes of the networks and the computational costs are indicated by the total number of their parameters and MACs per frame. It should, however, be noted that there is a small difference of these values when using the unified or the separate codebooks: in the latter case, there is one additional entry, so the output layer of the classification networks needs to be modified accordingly. However, the difference caused by the choice of the codebook is negligible compared to the total size and complexity of the networks. Therefore, we report here the network size and MACs taking the unified codebook as example. As with the baseline system summarised in Table 1, the C-DNN with four FC hidden layers with 73 units in each layer has 29,820 parameters and performs 27,448 MACs per frame.

For the GRU classifier, we took one single layer of GRU with 62 nodes, which yielded a network with 19,656 parameters and 19,220 MACs per frame, as shown in Table 2.

These two classification networks were trained on the negative log-likelihood (NLL) loss function. Since the training set was imbalanced, the NLL loss was further weighted by the inverse of the normalised distribution of the codebook entries on the training set. The learning rate was 0.001 for all networks. It was shown in Reference [11] that envelopes estimated in an MMSE manner have an advantage over their MAP counterparts. Thus, we adopted the MMSE approach for all classifiers.

Detailed parameters of the CRNN that predicts envelope coefficients from noisy LPS and the LSA gain are listed in Table 3. The regression network was trained by the MSE loss between the network prediction and the clean reference envelope coefficients.

### 4.2. Quality Measures

We evaluated the quality of the processed signal through four metrics from different perspectives. The speech quality was measured based on the wide-band perceptual evaluation of speech quality (WB-PESQ) [28]. We used the mean opinion score–listening quality objective (MOS-LQO) scores whose range fell between 1.04 and 4.64 for the evaluation. In the following text, we denote WB-PESQ MOS-LQO as PESQ in shorthand. The second metric was short-time objective intelligibility (STOI) [9]. STOI indicates the speech intelligibility as a value between 0 (incomprehensible) and 1 (perfect intelligibility). A higher score is preferred on both metrics.

Apart from these two widely used metrics, we also employed the white-box approach [5] to separately benchmark noise suppression and signal distortion. To this end, the final gain function estimation ${\tilde{G}}_{l}\left(l\right)$ was applied to the speech and noise component of the noisy input in order to obtain the filtered components, respectively:(9)$$s^{\prime }=\mathrm{iSTFT}\left\{{\tilde{G}}_{l}\left(l\right)\cdot S_{l}\left(l\right)\right\}\;,$$
and
(10)$$v^{\prime }=\mathrm{iSTFT}\left\{{\tilde{G}}_{l}\left(l\right)\cdot V_{l}\left(l\right)\right\}\;.$$
then, noise attenuation (NA)—the metric that measures the noise reduction ability—was given by:(11)$$\mathrm{NA}=10\log_{10}\left[\frac{1}{L}\sum_{l=0}^{L-1}\frac{\sum_{k=0}^{T-1}v{\left(k+lT\right)}^{2}}{\sum_{k=0}^{T-1}v^{\prime }{\left(k+lT\right)}^{2}}\right]\;,$$
where *T* is the frame length.

Similarly, the introduced signal distortion was measured by the segmental speech-to-speech-distortion ratio (SSDR) as:(12)$$\mathrm{SSDR}=\frac{1}{\left|\right|L_{1}\left|\right|}\sum_{l\in L_{1}}10\log_{10}\left\{\frac{\sum_{k=0}^{T-1}s{\left(k+lT\right)}^{2}}{\sum_{k=0}^{T-1}{\left[s\left(k+lT\right)-s^{\prime }\left(k+lT\right)\right]}^{2}}\right\}\;,$$
where $L_{1}$ is the set of speech active frames, and ||·|| is the cardinality of the set.

### 4.3. Oracle Test Results

First of all, we investigated the optimal speech envelope representations by using the oracle tests. In these tests, the ground truth of the envelope codebook entries are available while the two-stage framework is maintained. In other words, the oracle tests demonstrate the upper bound of the envelope enhancement method in the given two-stage framework. The performance difference among these systems depends, then, purely on the adequacy of representations and the codebook generation methods in this task. Therefore, we can choose the optimal feature based on the oracle test results. The features were examined in the following aspects: (a) the original codebook whose corpus was manually separated into two categories (dubbed ‘separate codebook’) vs. the proposed codebook that was created from all available materials (‘unified codebook’); (b) LPCC vs. CC as the speech envelope representation. Apart from the preliminary denoised results (LSA), one extra baseline used was the oracle regression method, which utilizes the unquantised clean envelope coefficients in the two-stage framework. Comparison of this oracle regression and the oracle codebook systems measures the distortion introduced by envelope quantisation. The evaluation results of these oracle tests are shown in Figure 5.

#### 4.3.1. Codebook Optimisation: Findings from Oracle Tests

In terms of the speech quality, the unified codebook has a clear advantage, and the gap between two codebooks increases with the input SNR. When the SNR exceeds 5dB, the separate codebook quantisation even degrades compared to the output from preliminary denoising. The separate codebook has a marginal advantage on STOI when the input SNR is low (≤0 dB). Yet, it is still questionable how much of this advantage can be replicated by a trained classifier under such low SNRs and in realistic settings. As the input SNR increases, the unified codebook begins to gain a small advantage. However, no method shows a significant improvement on STOI compared to the noisy input signal when SNR >5dB. This is somewhat expected, as the intelligibility of the noisy input signal also increases at higher SNR and, especially above 5dB, the intelligibility is quite high (close to 0.9, signifying almost perfect intelligibility). With the white-box decomposition of different methods on the noise and speech component, it is clear that the unified codebook mainly improves the noise reduction ability of the system. Given the fact that the major difference between the two codebooks lies in the envelopes of low-energy frames, it is understandable that its influence on the speech distortion metric is small.

In general, the evaluation results of the oracle tests indicate that the unified codebook is more suitable for envelope enhancement than the baseline separate codebook. This improvement from the baseline is in line with our analysis of the distribution of the codebook entries of clean speech: a direct clustering of all frames is enough to create the codebook, because the envelope patterns of speech-inactive frames show low overlap with their speech-active counterparts. It is also beneficial to have a finer quantisation of the speech-inactive frames, even though their energy levels are low.

#### 4.3.2. Feature Representation

No matter which codebook is chosen (unified or separate), CC has a consistent advantage over LPCC in terms of both metrics most of the time. LPCCs only provide a marginally higher boost to STOI when the input SNR is −5dB. It is clear that CC is more appropriate than LPCC for this task. When the input SNR is high, the choice of codebook actually plays a more important role in the final speech quality. The NA-SSDR decomposition indicates that both features perform similarly in terms of noise reduction when the input SNR is low. CCs show a clear advantage in both components over LPCCs when using the separate codebook. When the codebook is generated in a unified way and the input SNR is higher than 0dB, there is a trade-off between noise reduction and signal distortion: CCs introduce less speech distortion while LPCCs suppress more noise. This divergence grows as the input SNR increases. Yet, it should be noted that the difference in noise reduction is smaller than the difference in signal distortion.

Based on the observations on the oracle tests, we can conclude that cepstral coefficients quantised by the unified codebook demonstrate the greatest potential for application in two-stage speech enhancement. Consequently, this is the feature set we shall use in the subsequent evaluations.

#### 4.3.3. Quantisation Error

Comparing the best oracle classification system (the CC-based unified codebook) with the oracle regression system, the major difference comes up at high SNRs on PESQ. It is interesting to note that a better envelope restoration provides more benefit if the input signal itself is of higher quality, which presumably comes from a clearer excitation signal. When the excitation signal is poorly structured, applying an improved envelope to it introduces vocoding noise. On the contrary, if the excitation signal is of good quality, a good envelope estimation can restore the underlying speech to a better degree. The quantisation makes basically no difference on STOI scores. From the PESQ and STOI scores, it is clear that even with perfect envelope estimation, there is still room for improvement in the two-stage framework with oracle envelope information, especially under low SNR conditions. Generally speaking, envelope enhancement only provides us with a substantial improvement in speech quality when the input SNR is high. This indicates that our two-stage framework is limited by other components in the system, e.g., the noise floor estimator and the quality of the initial estimate.

### 4.4. Practical System Evaluation Results

Next, we evaluate the trained speech envelope estimators on the same test set. Apart from the optimal feature set decided in Section 4.3—the unified codebook based on cepstral coefficients (CCs)—we also evaluate the classifiers using CCs quantised by the separate codebook to verify our conclusions from the oracle test results.

#### 4.4.1. Impact of Separate Codebook

First, we investigated the impact of the codebook entry assignment errors on a practical system when using the separate codebook. Figure 6 compares the GRU classifier based on the separate and the unified codebook against three baselines: (i) the results from the preliminary denoising; (ii) oracle assignment of codebook entry based on the separate codebook; and (iii) oracle assignment of codebook entry based on the *unified* codebook.

Using envelope enhancement (with separate or unified codebooks) yielded a consistent improvement over the preliminary denoised signals. However, an interesting observation is that, when using the separate codebook, the GRU performs *better* than the corresponding oracle system when SNR is higher than 5dB. This is true for both metrics and is more obvious with PESQ. However, this seems somewhat unsettling given that the oracle assignment should represent the upper bound. This peculiar phenomenon only makes sense when taking our previous analysis on the separate codebook into account: the energy-based partition of the clean speech signals blurs the differences among the low-energy frames and excessively simplifies many frames into a flat envelope—both of which lead to degradation of the subsequent enhancement stage. Our statement can be circumstantially verified by this extra improvement from the GRU. Compared with the LSA baseline, the improvement by the GRU classifier on both metrics indicates that the network successfully learns to map the distorted envelopes to the pre-determined templates, although there is the noise of assignment errors in the training data due to the extra energy criterion. When the input SNR is high enough, the network is able to ‘correct’ the silent ‘ground truth’ labels to a better estimate from the speech envelope templates. It is this correcting that makes the trained classifier a better estimator than the oracle entry assignment for the separate codebook, which would force a flat spectrum on such frames. Note, however, that this GRU estimation is never as good as the oracle system using the *unified* codebook, which assigns all of the frames solely by their envelope coefficients. Further, the unified-codebook-based GRU classifier is never better than this oracle system, either. This further provides empirical validation of our analysis regarding the benefit of the unified codebook vs. the separate codebook, and the selection of the oracle assignment as the upper bound.

In order to further verify our hypothesis, we plot the envelope estimation cepstral MSE error of the three systems in Figure 6c. When using the separate codebook, the GRU-classifier gains an advantage over the oracle system from 5dB onward, which is in line with other objective metrics. This, again, provides us with a reason to choose the unified codebook over the separate one in the practical system.

#### 4.4.2. Comparative Benchmark against DNN Baseline

Having established the superiority of the CC-based feature representation and the use of a unified codebook, we now focus on benchmarking the performance of this envelope estimator against the DNN classifier baseline [11] and the CRNN-based regression approach described in Section 3.3.3 which directly predicts the envelope coefficients. The evaluation results are shown in Figure 7, where the results of the preliminary denoising are also included in order to better interpret the additional benefit provided by enhancing the envelope in the two-stage framework.

Generally speaking, all three methods show consistent improvement from the preliminary denoised signals. In terms of the speech quality, two classifier networks (DNN and GRU) perform similarly. GRU has a minor advantage when the input SNR is low, while DNN scores slightly higher when SNR is high. The CRNN regression network, however, shows a consistent improvement over the two classifiers at even lower computational cost.

If we take a closer look at the NA-SSDR decomposition, it can be observed that, at low SNRs, the GRU and the CRNN architectures better preserve the target signal (SSDR is 1–2 dB more) compared to the DNN architecture, whereas the DNN architecture has better noise attenuation (up to ∼1.5 dB more) here. The better signal preservation could be due to the temporal modeling capability introduced by the recurrent layers, which is absent in the DNN architecture. In most cases, the regression network has better signal preservation than the codebook methods, the reason for which could be the inherent limitation due to quantisation in the latter methods. The choice between the GRU codebook approach and the CRNN regression approach is a trade-off between noise reduction and speech distortion.

All of the trained networks demonstrate only minor differences on STOI compared to the noisy signals. Nevertheless, compared with the preliminary denoising results, the boost to STOI is consistent. When SNR is at 0dB or 5dB, the networks marginally benefit the speech intelligibility.

In Figure 8, we provide two sets of samples to illustrate the performance of the proposed envelope estimators. Compared to the clean reference envelopes, it is clear that the CRNN better preserves the speech details, which gives us an intuitive impression of the performance difference of the two networks: when the speech is better estimated in the initial stage, the regression network provides a more detailed structure of the envelope, whereas the classifier constrained by the codebook seems more beneficial when the speech is unclear. This trade-off can also be observed from the audio samples: https://aspireugent.github.io/speech-envelope-estimation/ (accessed on: 16 April 2023).

Envelope estimation can affect the excitation signal generated by CEM. Therefore, the optimal combination of two methods requires further investigation.

## 5. Conclusions

In this paper, we investigated and optimised the cepstral envelope estimation for speech enhancement using the two-stage framework. Through oracle tests, we conclusively demonstrated that cepstral coefficients provide a better envelope representation compared to linear prediction cepstral coefficients. Furthermore, the manual division of the speech/non-speech frames for codebook creation was shown to be unnecessary and even *detrimental* to the system performance. Using the optimal envelope feature representation, the GRU-based classifier achieved better performance than the baseline feedforward DNN-based classifier. This performance improvement was, additionally, obtained with fewer parameters and lower computational cost. Envelope estimation could be further improved by performing a regression onto the envelope coefficients instead of utilising a codebook-based template. The CRNN network designed for the regression took the noisy input spectrum and initial gain function estimate as input and performed better with a lower computational cost in comparison with the codebook-based estimator. Compared to the initial speech estimate (preliminary denoising), all of the evaluated methods brought benefits to the quality of the enhanced signal without reducing the intelligibility.

More importantly, the oracle tests revealed that the fundamental shortcoming of the two-stage framework lay not in the envelope estimation, but in limitations resulting from other components, such as the noise floor estimate and the statistical-model-based gain function, which performed poorly in very dynamic noise conditions.

Given a better initial estimate of the underlying speech signal, the proposed envelope estimators could be integrated into the signal processing pipeline in post-processing or as a second neural network focusing on the envelope estimation.

In summary, if the goal is to have improved single-microphone noise suppression within an interpretable, controllable, low-cost framework, then the work presented in this paper may be a good option. On the other hand, end-to-end enhancement can yield better noise suppression and speech quality, but at the cost of higher computational expense, poorer interpretability, and lack of control possibilities. 

## Figures and Tables

**Figure 1 sensors-23-06438-f001:**
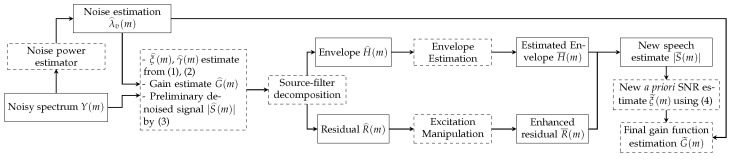
Block diagram of the gain function calculation in two-stage noise reduction. Dashed boxes represent manipulation blocks whereas solid rectangular boxes indicate data contained. Please note that all terms are in the STFT domain, where the frame index *l* has been dropped for conciseness.

**Figure 2 sensors-23-06438-f002:**
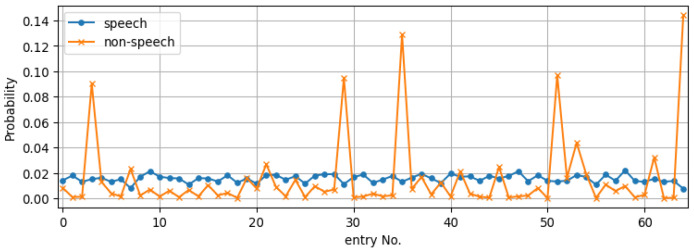
Distribution of speech/non-speech frames where the speech codebook entries are obtained from the speech active frames and the non-speech frames are idealised by a single entry with a flat envelope. We term this the *separate* codebook. Note, however, that when assigning non-speech frames on this codebook using the cepstral distance, they often correspond to codebook entries of speech-active frames.

**Figure 3 sensors-23-06438-f003:**
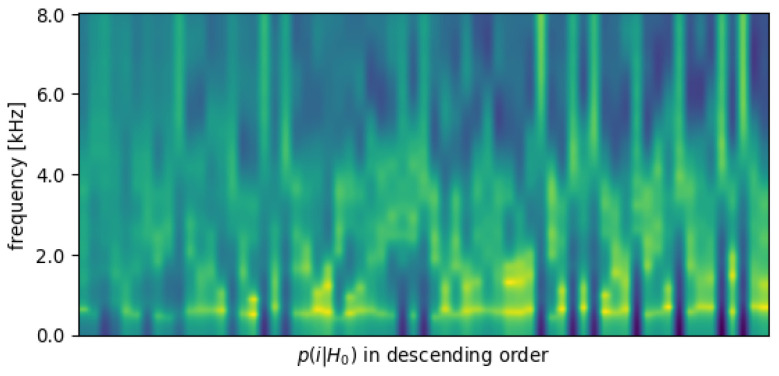
Envelope templates in the separate codebook, arranged by the posterior distribution of the codebook entries on non-speech frames ($p(i|H_{0})$). The envelope with the highest possibility of being non-speech is on the left.

**Figure 4 sensors-23-06438-f004:**
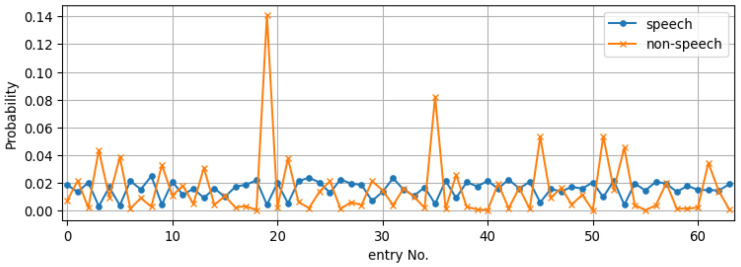
Entry distribution of speech/non-speech frames on the unified codebook. Here, the distributions of the speech and non-speech frames across the codebook entries seem relatively mutually exclusive—speech-active frames rarely correspond to codebook entries where non-speech frames show a high probability of association.

**Figure 5 sensors-23-06438-f005:**
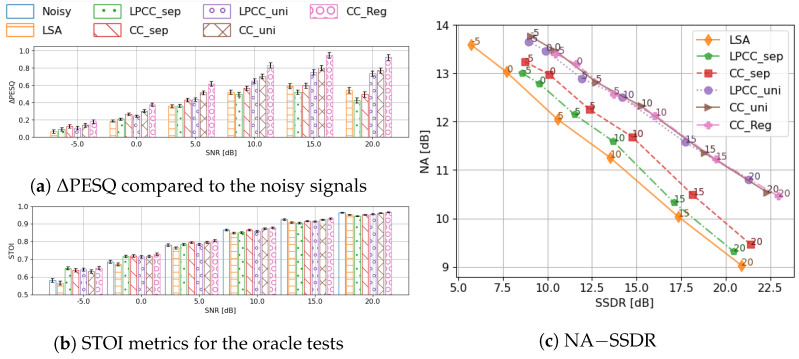
The oracle system evaluated by ΔPESQ, STOI, and NA−SSDR, and grouped by input SNRs. For the bar−plots, the 95% confidence interval is given by the error bars. LSA: the preliminary denoising output; _sep: using separate codebook; _uni: using unified codebook; _Reg: the regression method.

**Figure 6 sensors-23-06438-f006:**
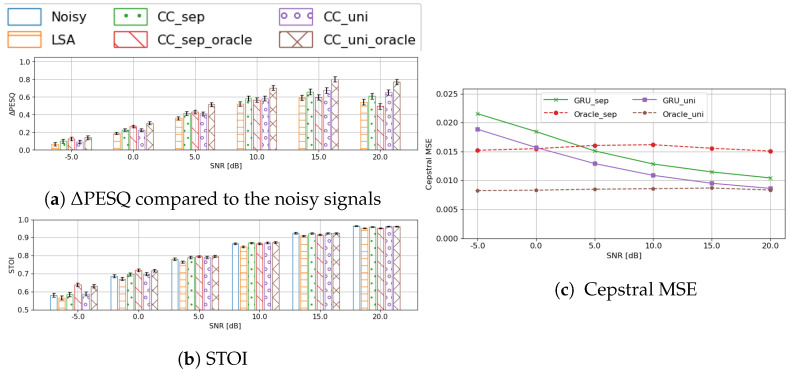
Comparison of the oracle systems and the GRU-classifier based on PESQ, STOI, and cepstral coefficient prediction MSE, grouped by input SNRs.

**Figure 7 sensors-23-06438-f007:**
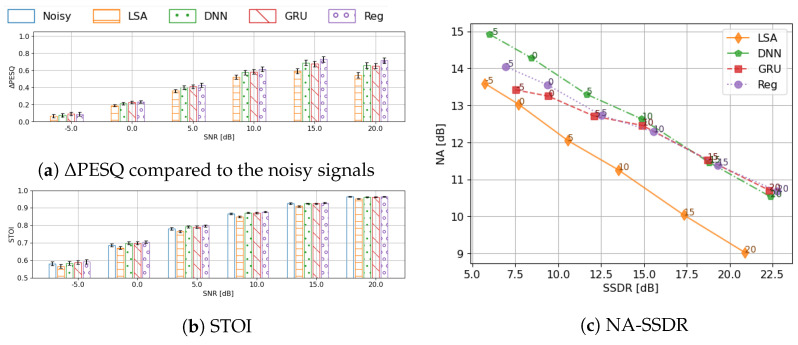
Comprehensive benchmark of the proposed systems against the DNN baseline in realistic settings. Performance is evaluated by ΔPESQ, STOI and NA−SSDR, grouped by input SNRs. The 95% confidence interval is given by the error bars in the bar−plots.

**Figure 8 sensors-23-06438-f008:**
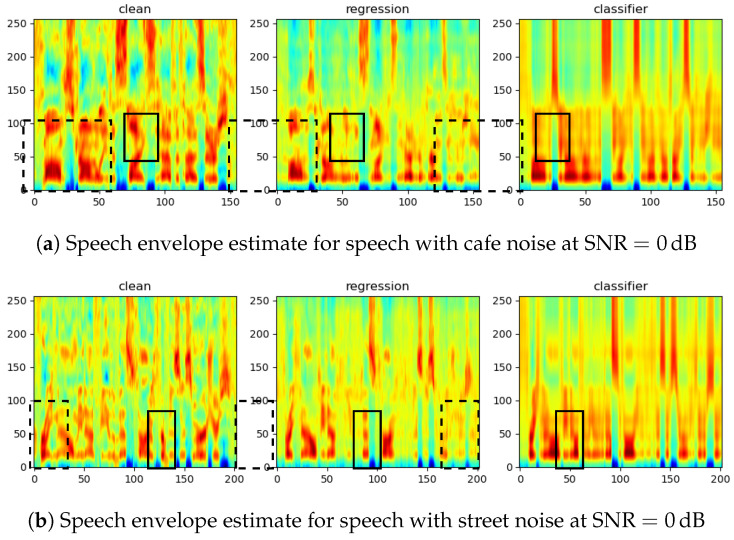
Comparison of the speech envelope estimate by different estimators. We highlight the regions where the CRNN-regression network estimates a more refined structure with dashed rectangles and the regions where the GRU-classifier shows an advantage with solid rectangles.

**Table 1 sensors-23-06438-t001:** Parameters of the Feedforward DNN classifier.

Input Size	#Hidden Layers	#Unit in the Hidden Layer
N=20	4	73
Default activation functions		Leaky ReLU, slope =0.03
Activation functions of the last layer		Sigmoid
Output normalisation		Softmax
Number of parameters		29,820
MACs per frame		27,448

**Table 2 sensors-23-06438-t002:** Parameters of the GRU classifier.

Input Size	#GRU Layers	#Unit in the GRU Layer
N=20	1	62
Activation functions		Sigmoid
Output normalisation		Softmax
Number of parameters		19,656
MACs per frame		19,220

**Table 3 sensors-23-06438-t003:** Parameters of the CRNN regression net.

Channels	4,8,8,1
Kernel size (Time = 1, Frequency)	3,3,3,1
Stride (Time = 1, Frequency)	2,2,1,1
Default activation functions	Leaky ReLU, slope =0.03
Number of parameters	4101
MACs per frame	11,044

## Data Availability

Not applicable.

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
