# Peer review of "Investigations on the Optimal Estimation of Speech Envelopes for the Two-Stage Speech Enhancement"

_sensors, 2023, doi:10.3390/s23146438_

Round 1
Reviewer 1 Report
With interest we were reading the paper. But we have the following comment:
1.The paper is quite readable. We miss the problem definition and a final conclusion if the goals are realized
2. The authors use a two stage framework which is hardly discussed
3. We noticed that a similar paper has been published on a Workshop (see 2), this should be clarified
4. we noticed that recent papers are not included, we noticed a paper 2 on brain computer with extensive literature study
1. 1. Investigations on the optimal estimation of speech envelopes ...
ugent.be
https://biblio.ugent.be › publication
door Y Song · 2021 — “Investigations on the Optimal Estimation of Speech Envelopes in Speech Enhancement.” In 2022 Speech in Noise Workshop, Abstracts, 42–42.
2. 2. Two stages of speech envelope tracking in human auditory cortex modulated by speech intelligibility
Get access ![]()
Na Xu, Baotian Zhao, Lu Luo, Kai Zhang, Xiaoqiu Shao, Guoming Luan, Qian Wang, Wenhan Hu, Qun Wang
Cerebral Cortex, Volume 33, Issue 5, 1 March 2023, Pages 2215–2228, https://doi.org/10.1093/cercor/bhac203
Reviewer 2 Report
This paper presents a method for speech enhancement.
The studied topic is interesting and also meaningful.
The paper still has some major problems.
The authors are suggested to revise the paper given the following comments.
As discussed in some studies (‘Blind quality assessment based on pseudo-reference image’, ‘Blind image quality estimation via distortion aggravation’, ), image/video quality is an important aspect of various intelligent systems. High-quality images/videos are important for the successful usage of these intelligent systems, while low-quality media may degrade the performance of these systems.
The authors may give some discussions on this aspect and the above mentioned works.
The authors mainly discuss speech/audio quality enhancement in this paper. However as discussed in the literatures, ‘Study of subjective and objective quality assessment of audio-visual signals’ audio and visual cues will fuse and together shape the overall experience.
The authors are suggested to give some discussions on this aspect and the above works.
The writing of the paper still needs improvement. The whole paper is suggested to be double-checked to remove all possible issues.
The writing of the paper still needs improvement.
Reviewer 3 Report
The authors investigate and optimize the cepstral envelope estimation for speech enhancement using the two-stage framework. Through oracle tests, the authors decide the most appropriate model structure for envelope estimation based on CRNN, such as input representation, no separation of the speech/non-speech frames for codebook creation and GRU-based classifier. The experiments show that, in the first stage, all evaluated methods bring benefits to the quality of the enhanced signal, without reducing the intelligibility; and the proposed model for regression provides the performance improvement comes at only a marginally higher computational cost compared to the codebook-based estimator.
The model design is sound verified by many previous studies and the authors own oracle tests. And the block diagram and charts of model component selection are well-organized and easy to understand.
In the experimental part, the authors conducted several ablation experiments, confirming that the model components used in this paper (Cepstral coefficients feature, no separation of the speech/non-speech frames, integrated codebooks and GRU-based classifier) have good denoising performance and low time complexity for the two-stage speech enhancement task.
It is commendable that the paper thoroughly explains each step of the model design concept and experimental phenomena, such as highlighting the advantages of integrated codebooks on voice quality and the limitations of discrete codebooks at low SNRs.
However, there are some questions about this paper:
1. Lack of consistency between symbol notations and literal interpretations on block diagrams, such as “posterior SNR estimates”.
2. When describing a codebook in Section 3.2, why the distributions of voice and non-voice frames across codebook entries appear relatively mutually exclusive.
3. Explain why the probability value significantly increases from the entry number 19 in Figure 4 in more detail.
The quality of English language in this paper is satisfactory. The thesis of speech enhancement is well-structured. Additionally, the paper is fluent-writing and there are no major grammatical errors.
Round 2
Reviewer 2 Report
Some of the previous concerns have been addressed.
However, the mentioned related works are not properly discussed in the revised paper.
The authors are suggested to discuss the related topics as well as the mentioned related works.
None
Author Response
Dear reviewers,
First of all, thank you for the valuable feedback, which helped us improve the readability of our work. The following references:
Blind quality assessment based on pseudo-reference image
Blind image quality estimation via distortion aggravation
Study of subjective and objective quality assessment of audio-visual signals
primarily focus on image quality assessment and/or the influence of visual-audio cues and their consistency on the overall experience in multi-media systems. Undoubtedly, these topics are interesting and important in the context of audio-visual systems. Nevertheless, our paper specifically focusses on audio enhancement (such as used in communication systems). Therefore, the discussions about visual cues are out of scope for this contribution – and also not our area of expertise. We cannot provide a comprehensive overview of audio-visual quality in this paper, and the related discussion would distract our readers’ attention from the core topic of this paper. In our opinion, neither are helpful to the flow and the readability of this work.
Again, as we explained in the previous round of rebuttal, those papers would be very interesting and relevant to our future work in audio-visual systems. However, for the purpose of the current paper, we kindly request your consideration in allowing us to skip these references for the moment.